# The Interrelation of Neurological and Psychological Symptoms of COVID-19: Risks and Remedies

**DOI:** 10.3390/jcm9082624

**Published:** 2020-08-13

**Authors:** Mohammad Nami, Bharathi S. Gadad, Li Chong, Usman Ghumman, Amogh Misra, Shrikanth S. Gadad, Dharmendra Kumar, George Perry, Samuel J. K. Abraham, K. S. Rao

**Affiliations:** 1Neuroscience Center, Instituto de Investigaciones Científicas y Servicios de Alta Tecnología (INDICASAT AIP), City of Knowledge 084301103, Panama; 2Department of Neuroscience, School of Advanced Medical Sciences and Technologies, Shiraz University of Medical Sciences, Shiraz 71348-14336, Iran; 3Dana Brain Health Institute, Iranian Neuroscience Society-Fars Chapter, Shiraz 71364-76172, Iran; 4Academy of Health, Senses Cultural Foundation, Sacramento, CA 66006, USA; 5Department of Cognitive Neuroscience, Institute for Cognitive Science Studies (ICSS), Pardis, Tehran 1658344575, Iran; 6Department of Psychiatry, Paul L. Foster School of Medicine, Texas Tech University Health Sciences Center at El Paso, El Paso, TX 79905, USA; tssrao2009@gmail.com; 7Beijing Zhongke Jianlan Biotechnology Co., Ltd., Beijing 100101, China; chongli@moon.ibp.ac.cn; 8Zhongke Jianlan International Medical Research Institute, Melbourne, Victoria 3000-3999, Australia; 9Institute of Biophysics, Chinese Academy of Sciences, Beijing 100101, China; 10Department of Biochemistry, University of Texas, Austin, TX 78712, USA; kjr5n2009@gmail.com; 11Center of Emphasis in Cancer, Department of Molecular and Translational Medicine, Paul L. Foster School of Medicine, Texas Tech University Health Sciences Center El Paso, El Paso, TX 79905, USA; shrikanth.gadad@ttuhsc.edu; 12Department of Molecular and Translational Medicine, Paul L. Foster School of Medicine, Texas Tech University Health Sciences Center El Paso, El Paso, TX 79905, USA; 13Graduate School of Biomedical Sciences, Texas Tech University Health Sciences Center El Paso, El Paso, TX 79905, USA; 14Greater Heights Holistic Psychiatry, Houston, TX 77008, USA; Drkumar.dharmendra@gmail.com; 15Semmes Foundation Distinguished University Chair in Neurobiology, The University of Texas, San Antonio, TX 78249, USA; george.perry@utsa.edu; 16School of Medicine, Yamanashi University, Chuo 409-3898, Japan; drsam@nichimail.jp

**Keywords:** mental health, COVID-19, neurological, psychological, inflammation

## Abstract

COVID-19 has catastrophically affected the world’s panoramic view of human well-being in terms of healthcare and management. With the increase in the number of cases worldwide, neurological symptoms and psychological illnesses from COVID-19 have increasingly upsurged. Mental health illness and affective disorders, including depression, obsessive-compulsive disorder, anxiety, phobia, and panic disorders, are highly impacted due to social distress. The COVID-19 pandemic not only affected people with pre-existing mental and affective illnesses, but also healthy individuals with anxiety, worrying, and panic symptoms, and fear conditioning. In addditon, the novel coronavirus is known to impact the central nervous system in the brain, resulting in severe and certain long-lasting neurological issues. Owing to the significance of neurological and psychological events, the present perspective has been an attempt to disseminate the impact of COVID-19 on neural injury through inflammation, and its interrelation with psychological symptoms. In this current review, we synthesize the literature to highlight the critical associations between SARS-CoV-2 infection and the nervous system, and mental health illness, and discuss potential mechanisms of neural injury through psycho-neuroimmunity.

## 1. Introduction

The current world has changed in all shapes since the emergence of the novel coronavirus. COVID-19 is a deadly virus and is also extremely contagious. The rapidly emerging COVID-19 pandemic has been taking a toll not only on overall health but also on the mental health of the general public, and especially those with previous history or episodes of mental illness [1]. A significant fraction of stress arises from the experience of monitoring ourselves or being monitored by others for the possible signs and symptoms of the disease [2]. Meanwhile, much of the mental health issues related to the COVID-19 pandemic stem from accumulating worries and concerns related to continued isolation, financial insecurity, contamination, and less-likelihood in returning to normality. Of note, the current state of health emergency has forced millions of people to stay socially distant from friends, family, colleagues, and even from their pets. Summing this up with the exponential fear, worrisome daily news, and fearful updates as well as the looming indefinite future, the rate of anxiety and stress- and fear-related disorders and depression seem to be on a steep rise. This is partly because we face a global issue, and as such, a global coalition is required to battle against COVID-19 [3].

In the latest pandemic time, it has become apparent that neurological and psychological involvement in COVID-19 are increasingly rising in the patient population. A subset of COVID patients is affected by neurological events such as headache, dizziness, or cerebrovascular symptoms [4,5,6,7]. The latest reports have also shown a sudden onset of anosmia and ageusia as an early sign of SARS-CoV-2 infection, suggesting that early neurological involvement might be highly relevant [4,5,7]. Currently, long-term neurological complications from COVID-19 infection are of great concern. Besides, the overarching issue is the situation of the socio-economic crisis and psychological distress rapidly occurring worldwide. Although social activities have been restricted in most countries, almost all movements were prohibited due to quarantine, leading to many psychological problems. The most critical consequences of the COVID-19 epidemic are of increased mental health issues, including stress, anxiety, and panic symptoms, and depression that has emerged increasingly [6,8]. The most common psychological symptoms are generalized fear and pervasive community anxiety, typically associated with disease outbreaks, with the escalation of new cases leading to increased anxiety and panic symptoms [6,8]. Overall, in this present review, we synthesize the literature to highlight the significant associations between SARS-CoV-2 infection and the nervous system, and mental health illness, and discuss potential mechanisms of neural injury through psychoneuroimmunity. Awareness of the possible psychoneurological manifestations in COVID-19 patients is of utmost importance for the management of potentially life-threatening and long-lasting psycho-neurological complications.

## 2. Key Challenges

The strong conditioned fear and anxiety about a disease potentially lead to often-overwhelming emotional dysregulation and mood changes both in adults and children [3]. Evidence has supported the notion that stress-related disorders not only leave an impact on overall mental health but also leverage neurocognitive predicaments. Indeed, stress is regarded as a significant yet under-appreciated factor in neurodegenerative diseases [9]. The critical challenge is that how we may cope with such growing anxiety differs in people based on their mental health background and the community or support system they live in [10].

The increased number of COVID-19 patients and shortage of staff and limited number of social interactions with physicians or healthcare professionals [11] led to a decline in face-to-face consultations and an increase in telemedicine. This, in turn, has caused elevations in stress levels, especially in individuals with severe mental health problems [12]. Moreover, because of COVID-19, all the elective surgical procedures have been postponed or canceled. They have caused immense stress in people that were scheduled for elective surgical procedures and those who require face-to-face consultations [13]. Some people including (1) older adults or those with a chronic health problem, (2) children and adolescents, (3) those who help with COVID-19 such as first-line physicians and healthcare professionals, and (4) individuals with underlying mental health issues, affective disorders, and substance abusers are expected to experience and may reveal more substantial distressing worries in the present crisis [14].

The above can, in turn, alter one’s affective health, cognitive fitness, sleeping, and eating patterns whereby maladaptive motivational structures such as the use of alcohol, smoking, and substance abuse tend to develop. As for the people with pre-existing mental illness, awareness about emerging or worsening symptoms becomes crucial. Individuals’ reciprocal care and support and the significance of family function in today’s crisis not only help people cope better with such anxiety but also make communities stronger against the COVID-19 physical and mental health burden [15,16,17]. Apart from all the recommended self-care measures for mental health and stress management, which we will briefly summarize here, the below points may also be of value where applicable. Such advice would include: (1) following the local public healthcare providers’ advice and refraining from ill-advised comments, (2) seeking mental health services online or teletherapy where applicable and possible, (3) requesting prescription refills for the next 90 days supply in case one is taking particular medicines and adherence to medication therapy where needed, and (4) raising awareness on self-care particularly in people of a higher risk profile [18]. This would per se increase the risk and subsequent worry that the disease would spread faster and take a higher toll.

## 3. Geographical Distribution of COVID-19

Globally, more than 16 million confirmed cases of COVID-19 have been reported. By 28 June 2020, so far, the number of cases stands at 10,243,859 cases with 504,410 deaths and 5,553,495 recovered worldwide; the United States has topped the list among all the countries in the world with total cases 2,596,537; 128,152 deaths; 1,081,437 recovered; 1,386,948 confirmed cases; and 15,816 are in critical condition (courtesy: www.worldometers.info/coronavirus) (Figure 1). With the increase in the number of cases in the United States and worldwide, the active cases in the United States have surpassed the numbers recovered, which is an alarming situation that needs to be addressed. The number of cases and deaths data were accessed from the WHO and CDC until 28 June 2020. However, the case fatality rate is low for SARS-CoV2 (3–4%) in comparison with SARS-CoV (9.6%), case fatality source: https://www.worldometers.info/coronavirus/coronavirus-death-rate/ Also the death rate compared to active confirmed cases is low. To succumb to the current situation, strict rules and regulations from the CDC must be followed to reduce the number of cases. However, with these numbers of cases reported, it is unclear how many of the COVID-19 subjects have secondary complications with neurological or psychosocial or mental health issues. It is also the need of the hour to screen for other illnesses that have been impacted by COVID-19.

## 4. Psychological and Neurological Symptoms of COVID-19

The major psychological symptoms among people include signs of anxiety, panic attacks, depression, and suicide [19,20]. To elaborate, the symptoms include persistent worrying or feeling overwhelmed by emotions; restlessness and irritability; sleep problems like insomnia or excessive sleeping; sweating, trembling, shortness of breath, or a sense of choking; and lack of interest, significant weight loss/gain, feelings of worthlessness or excessive guilt, and repeated thoughts of death or suicide [6,19,20]. In the latest report from Ellul et al. (2020), it was observed that in patients with COVID-19 with a neurological or psychiatric disease reported for over three weeks, the majority of them had an altered mental status, which included encephalopathy, with a neuropsychiatric diagnosis, including psychosis, neurocognitive syndrome, and affective disorders [7]. Notably, patients had a cerebrovascular event, ischemic strokes, intracerebral hemorrhages, CNS vasculitis, and other cerebrovascular events [6,7]. Hence, it is highly essential to understand the underlying biological mechanism on how these psychoneurological events are affected by COVID-19.

## 5. Psychological Impact from COVID-19

In recent times, the COVID-19 pandemic has immensely led to an epidemiological and psychological crisis. With the increase in the days of lockdown worldwide, living in isolation, changes in our daily lives, job loss, financial hardship, and death of loved ones have tremendously impacted on the potential to affect mental health-related issues [19,20]. During the time of social distancing, psychological symptoms and mental health issues have increased around the world.

In a recent review published in Lancet recently by Brooks et al. (2020), a study comparing post-traumatic stress symptoms in both parents and children quarantined vs. not quarantined found that the mean post-traumatic stress scores were 4-fold higher in children who had been isolated than in those who were not isolated [21]. A total of 28% (27 of 98) of parents quarantined in this study reported revealed symptoms of a trauma-related mental health disorder. All other quantitative studies were based on survey data, and reported a high prevalence of signs of psychological distress and disease [21]. So far, the studies published in the literature are high in general psychological symptoms, emotional disturbance, depression, stress, low mood, irritability, insomnia, post-traumatic stress symptoms, anger, and emotional exhaustion [21].

A study conducted by Pappa et al. (2020) was based on the protocols registered on PROSPERO which is based on the data pooled using random-effects meta-analyses to study the prevalence of specific mood-related issues [20]. The PROSPERO study was conducted on healthcare professionals [20], where a total number of 33,062 participants with 13 studies were included for the meta-analyses. In 12 studies, anxiety was assessed with a prevalence rate of 32%, and major depression in 10 studies with an incidence rate of 22.8% [20,22,23]. Further, female healthcare professionals revealed higher rates of mental health-related symptoms compared to male healthcare workers. Furthermore, insomnia prevalence was estimated at 38.9%, suggesting that sleep disturbances were observed to be a significant issue [20,22,23]. Based on these pieces of evidence, healthcare professionals and others are experiencing mental health issues, including sleep disturbances, during this outbreak. Hence, it is highly essential to launch new ways of interventions through counseling, social interactions, and psychotherapy through “telemedicine” under these uncertain pandemic conditions.

## 6. Neurological Effects of COVID-19

Though the consequences of COVID-19 have led to mental health issues, the brain biotypes and brain or the neural taxonomy are likewise affected. Even though COVID-19 mainly affects the respiratory system, recent studies have shown its direct involvement in the central nervous system (C.N.S.). A study done in Wuhan, China showed that almost around 40% of COVID-19 patients experienced neurological symptoms. These were further divided into C.N.S. symptoms, peripheral nervous system (PNS) signs, and musculoskeletal symptoms. COVID-19 patients with mild neurological symptoms complained of headache and dizziness, and more severe patients suffered from acute cerebrovascular diseases and consciousness impairment [24].

Additionally, reports have indicated olfactory and taste disorders in patients with COVID-19 [25]. COVID-19 is currently found to be affecting the neuroendocrine-immune system and heavily suppressing it [26]. The neuroendocrine-immune system has been said to be involved in stress and coping strategies. This might be one of the reasons for having excessive stress. Another reason for elevated amounts of stress might be because periods of isolation and lack of social contact have been linked to causing high psychological distress and stress-related disorders [27]. Older patients with underlying conditions have been reported with more severe neurological symptoms [24]. Of note, COVID-19 patients in Beijing Ditan Hospital were found to have the presence of SARS-CoV-2 in their cerebrospinal fluid, which was confirmed by genome sequencing, which led to the presentation of viral encephalitis [28].

The mechanism of action of COVID-19 and its neurological effects are currently under study. The C.N.S. is protected from viruses with its multilayer barriers and its immune response system. However, different viruses can affect the brain through a variety of mechanisms. Some proposed mechanisms by which the virus can cause infection include direct brain injury, hypoxic damage, upregulated angiotensin-converting enzyme 2 (ACE-2) receptors, and immune insufficiency, which can lead to toxic, infectious encephalopathy, viral encephalitis, and even acute cerebrovascular disease [28]. The viruses have been found to cause direct brain injury through different mechanisms [29] including via blood circulation where the virus is released into the blood, causing an increase in the penetrability of the blood–brain barrier that leads to the virus entering the brain which causes encephalitis [30]. Some viruses can also direct damage to the brain by involving the sensory or motor nerve endings [31].

It has been proposed that coronavirus causes its neurological symptoms via hypoxia. Since the virus primarily causes respiratory symptoms including shortness of breath and lack of oxygen in the lungs and consequent anaerobic metabolism in the brain, this can lead to brain injury displayed by brain swelling, interstitial edema, or cerebral vasodilation, etc. [32,33]. Thus, hypoxia resulting from COVID-19 infection can result in neurological symptoms. COVID-19 also has the potential to cause hyperinflammation through cytokine storm syndrome [34]. A retrospective study on 150 COVID-19 patients, which was carried out in Wuhan, China, revealed a significant increase in ferritin levels as well as IL-6 levels. Such findings were shown to be positively correlated with the fatality rates in these patients. This would point towards the role of inflammation in multiple COVID-19 symptoms, including its often-devastating neurological issues [35]. Previously, in a mice model, coronavirus was shown to potentially infect nasal cells from where it gets transported to the olfactory bulb before affecting the brain. The removal of the olfactory bulb caused a delay in the transmission of the virus to the mice’s brain [32,33]. Recently in Italy, a study reported that in COVID-19 cases, self-reported olfactory and taste disorders were found [25]. Despite such findings in the acute phase, long-term effects have yet to be studied. Another potential mechanism of COVID-19 causing brain injury is thought to be through ACE-2. ACE-2 receptors have been found in glial cells and neurons, which are a potential target of COVID-19. The attachment of the virus to these receptors facilitates the entry of the virus to the brain and resultant brain injury [36].

There is a crashing wave of neuropsychiatric sequelae of COVID-19. The major issues addressed related to COVID-19 are the neuropsychiatric symptoms and understanding the underlying immunological mechanisms. According to Julie Helms et al. (2020), 8 of the 58 COVID-19 patients (14%) admitted in the intensive care unit showed neurological and neuromuscular blockade [37]. Further, RT-PCR assays on the nasopharyngeal samples on all 58 patients revealed from subjects who were positive for acute respiratory syndrome COV2-SARS that seven developed ischemic attack, epilepsy, and mild cognitive impairment [37]. Further, since the impact of COVID-19 and mental health issues has recently been raised worldwide, it is essential to understand whether the rate or the association of suicidality has changed. It would, therefore, be interesting to follow the psychoneuroimmunology of COVID-19 subjects about the extent of mental health issues and the presence of suicidality. As one could speculate, the virus might lead to the immune response in vulnerable subtypes of the mood disorder population, as it was recently observed in cytomegalovirus [38].

## 7. Psychoneuroimmunity of COVID-19

One of the strong hypotheses on COVID19 and neuro-psychoimmunology is the strong interlink between changes in the cytokines and interleukins across the immune system [38]. Further studies have demonstrated the lack of naïve T cells, an increase in the senescent population of T lymphocytes, and shortening of telomeres in major depression [39]. A plausible approach to explore the association between the virus and the immune system function in major depression and other mood-related disorders is to check their immune response to the antigenic substance injected, such as endotoxin. The outbreak of a viral infection would be a great challenge, yet an opportunity to investigate the significance of an acute immune challenge on different domains of the psychopathology of mood disorders.

A study published in JAMA Neurology by Ling Mao et al. (2020) found that among 214 COVID-19 patients with a mean age of 52.7 [15.5] years, 126 patients had a non-severe infection, while 88 patients developed severe illness based on the respiratory infections [24]. Moreover, out of 214, 78 patients had neurological manifestations. Compared to non-severe infections, the severe ones were all from the elderly population and had underlying medical conditions, especially hypertension. COVID-19 patients with more severe infection had neurologic conditions such as acute cerebrovascular diseases, impaired consciousness, and musculoskeletal symptoms [24]. Furthermore, they observed that COVID-19 patients with severe infection had higher D-dimer levels compared to non-severe infection patients, suggesting the reason behind an increased risk for cardiovascular diseases [24]. Besides, patients with musculoskeletal symptoms were found to have elevated creatine kinase levels (400 U/L), and higher creatine kinase and lactate dehydrogenase levels than those without such symptoms [24]. Additionally, three patients had higher neutrophil counts, lower lymphocyte counts, higher C-reactive protein, and higher-D-dimer levels. The COVID-19 patients who already developed musculoskeletal symptoms also had multiorgan damage, including severe liver damage with increased lactate dehydrogenase, alanine aminotransferase, aspartate aminotransferase, blood urea nitrogen, and creatinine levels [24]. Hence, the infection-mediated immune response might have possibly caused abnormalities in the CNS. secondary to the upsurged interleukins. Patients with severe mental illness are bound to neglect the prevention of infection due to cognitive decline. The reduced physical activity due to the anxiety, fear of infection, and negative symptoms further leads to dysfunctional immunity [40]. However, patients free from COVID-19 infection are also psychologically impacted by the COVID-19 pandemic. Psychiatric morbidity is a paramount concern, as the virus may affect the CNS and provoke systemic inflammation [41]. A recent paper by Conti et al. (2020) reported that COVID-19 infection triggers the release of pro-inflammatory cytokines, including interleukin (I.L.)-1b and IL-6. SARS-COV2 is neurotropic and hence can invade nervous tissues and impede immune-functioning macrophages, microglia, or astrocytes in the CNS [42]. A neurotropic virus might activate microglial cells and induce a pro-inflammatory state [17]. The serum level of Interleukin-6, an essential member of the cytokine storm, is positively correlated with the severity of COVID-2019 symptoms [41]. Additionally, experiments have confirmed that primary glial cells cultured in vitro secrete a large number of inflammatory factors such as IL-6, IL-12, IL-15, and tumor necrosis factor-alpha (TNF-α) after being infected with CoV [32]. Furthermore, the hyperactivation of immune cells in the brain would eventually result in chronic inflammation and brain damage [28].

Bo Diao et al. reviewed the counts of total T cells, CD4+, CD8+ T-cell subsets, and serum cytokine concentrations retrospectively from 522 in-patients from two hospitals in Wuhan as well as 40 healthy controls [43]. Moreover, they further checked the expression of T-cell exhaustion markers programmed death-1 (PD-1), and T-cell immunoglobulin and mucin-domain containing-3 (Tim-3) were measured by flow cytometry in the peripheral blood of 14 COVID-19 cases. Based on their investigation, the T-cell count was found to be significantly reduced in COVID-19 patients, and the surviving T-cells were functionally exhausted [43]. Patients who were not at the ICU had a total T-cells, CD8+ T-cells, and CD4+ T-cells count lower than 800/μL, 300/μL, and 400/μL, respectively. This has mandated a continued aggressive intervention even in the immediate absence of more severe symptoms to prevent further deterioration in their condition [43]. There are also reports from the post-mortem pathology of COVID-19 patients who died in the U.S., showing the infiltration of CD-8+ T-cells within the alveoli [29].

Given the above, the cytokine storm created by an imbalance between CD-4+ and CD-8+ T-cells might be the plausible cause of acute respiratory distress syndrome in COVID-19 patients. Figure 2 depicts the viral and host factors that influence the pathogenesis of SARS-CoV-2. Physical and biopsychosocial and psychoneurooimmune effects impacted by the virus can be improved by a healthy lifestyle, exercise, a balanced diet, staying connected with family and loved ones using telecommunication or internet, and maintaining quality sleep. Hence, in a nutshell, it is highly imperative to build psychosocial resilience to enhance psychoneuroimmunity against the virus [40].

So far, clinical observations have substantiated that patients with COVD-19 may portray a variety of neurological signs and symptoms. Milder symptoms include headache and nausea, while in more severe cases, fatal encephalitis may develop. Meanwhile, older patients and those with underlying conditions are at higher risk for developing neurological symptoms. Of note, anosmia and ageusia are now known to be common early symptoms which people need to be aware of. The long-term effects of COVID-19 are yet to be researched. Therefore, only more long-term research will answer whether these neurological deficits are reversible, recurring, or permanent.

## 8. Advisable Remedies

**A.** **Change in lifestyle and coping to overcome psychological distress:** In today’s unprecedented health-related crisis, there are many tips and advice aimed to bring mental relief to people. It is important to acknowledge the anxiety as feelings, considerations, and reactions; exercise as a classic anxiety reduction strategy; staying physically distant but not socially distant; maintaining sleeping habits; and that meditative practice and a healthy diet might boost the immune system [44]. This will eventually improve a patient’s social and occupational functioning. Providing proper instructions and educating about the importance of mental health and the use of different modalities, including lifestyle changes, counseling, and psychotropic medications, can be beneficial. However, most people are trying to cope well, but it is highly important to identify early those individuals who are at risk for psychological morbidities such as younger individuals; addressing the emotional psychological and psychosocial responses in a timely manner; and establishing enhancing social support networks that can buffer against distress to combat negative psychological responses.**B.** **Psycho-education, training, and counseling:** The mental healthcare system should be persistent in providing necessary psychosocial treatments. This can happen only through obstinate uninterrupted ambulatory care by protecting the psychiatrists, psychiatry, psychosocial healthcare workers, and staff. Patients should be well informed on how and where to seek help for COVID-19 infection and their mental illness relentlessly. Over and above, stress management measures such as mindfulness meditative training, relaxation techniques, yoga, praying, and similar practices will assist the mindful presence with the power of silence. With regards to meditation, in a recent paper, a unique protocol was proposed to conquer worry in the age of COVID-19, offering some geometric meditation techniques, along with a link to a video clip file dedicated to each method available free for public use [45]. We are already working on developing and validating an android-based smartphone app in three languages (Persian, English, and Spanish) aiming to assist users in conquering their distressing worries in today’s life.

With everything that has changed with society as a result of the pandemic, there is a great need to develop more “wellness groups” to healthcare workers who are working tirelessly to treat patients during the current pandemic. These groups have also sparked a lot of conversation around where future needs may arise, as COVID-19 will undoubtedly be a large issue for the next 12–18 months. These groups and resources could also be adapted for other “frontline” essential workers, such as those in the food industry and delivery, as they may also experience similar stressors as healthcare workers. When one is responding to COVID-19 and is quarantined, mental predicaments become much harder. Even upon the release from quarantine, one might experience mixed feelings, including intense fear about his/her own and loved ones’ health. Further, other symptoms such as sadness and irritation because a friend or loved one might have contracted COVID-19 can be frustrating. Other emotional and mental changes often result from the guilt of not being able to interact with other people, complete tasks, and take duties during the quarantine period. Furthermore, the financial and social burden of the issue adds to the mental health burden of the disease [1]. In such an instance, one needs to take time for himself/herself and his/her family to recover from responding to the pandemic. It would also be advised that individuals take a break from media coverage of COVID-19 and seek community assistance instead [1].

## 9. Conclusions

While the mechanisms behind the COVID-19 disease burden are being studied, it is now clear that patients suffering from diseases may develop a variety of psychoneurological signs and symptoms. As the numbers of COVID-19 cases continue to rise worldwide, there is an increasing number of studies that have reported psychoneurological symptoms, with the latest reports suggesting that COVID-19 patients suffer from Guillain–Barré syndrome and other long-lasting neurological complications. Studies published mainly from China and France have also reported the significance of neurological and mental health disorders in COVID-19 patients. According to these reports, up to 36% of patients have demonstrated psychoneurological symptoms. As such, the neurological and subsequent neuropsychiatric burden of the disease would require even further attention in our today’s clinical practice against the virus. COVID-19 can potentially affect anyone, regardless of age, gender, and ethnicity.

Meanwhile, when someone has had mental traumas or experienced mental or long-term physical illness, or when an elderly finds himself/herself more vulnerable to the effects of coronavirus, the distressing worry turns to be off the chart. Thus, one needs to develop skills and awareness instead of making assumptions while acknowledging stress and managing it. This pandemic will be expected to continue reshaping the patient–physician relationship with the emergence of telemedicine. The pandemic would also continue to reinforce the global healthcare system to avoid further unpreparedness for similar crises. Over and above, public awareness campaigns and academic efforts need to be synergized to help people refrain from judging who is responsible for the virus spread instead of assisting them with how to stay mentally and physically safe by adequately following the outbreak updates from trusted resources. Increased screening of mental health symptoms during primary care or other medical specialty visits, and offering mental health services with the slightest degree of suspicion, can decrease the mental health burden.

In most cases, the COVID-19 related stress in a person without significant past psychiatric history can be diagnosed as an adjustment disorder. Adjustment disorder can be treated with counseling and minimal use of psychotropic medications. On the other hand, patients with pre-existing or chronic psychiatric illnesses may decompensate due to increased stress related to COVID-19. This patient population may benefit from more frequent mental healthcare visits, even if this can be provided via telemedicine. Increased visits with their mental health provider can ensure medication compliance and early detection of any relapse of their psychiatric symptoms. In specific cases, the physician may consider more liberal but very brief use of anxiety and sleep medications. The decision to use medication should be based upon the most recent evidence-based guidelines.

An accumulating body of recent evidence proposes that anxiety, depressive, and psychotic symptoms are all likely to worsen during extreme COVID-induced stress and social disruption. Moreover, patients will be at increased risk of relapse or recurrence of affective and psychotic illness. As such, it will be important when deciding on the best management plan (non-pharmacological/pharmacological) to consider all the relevant factors, including risk to self and others. It is important to understand the difference between short- and long-term use of psychotropic medications, and also to clarify the myths related to the addiction potential of all psychotropic medications. Although many psychiatric medicines are tightly regulated and prescribed only for long-term mental illnesses, it may be necessary for the governments to ease up prescription refill regulations. Teletherapy and online consultations with e-prescriptions would allow ease of access to the prescribed medications without referring to the mental healthcare provider in person. Perhaps in many occurrences, designated pharmacies might collect e-prescriptions plus the related contact info of the patients for home-delivery medicines. Notwithstanding the above, possible untoward or side-effects of such medication need to be noted. In other words, upon treatment with commonly prescribed psychotropic drugs, careful consideration should be given to whether now is the best time to commence, withdraw, or change patients from antidepressant, anxiolytic, or antipsychotic medications. For instance, in patients who receive ongoing treatment with benzodiazepines, the potential for tolerance and dependence needs to be considered. With regards to lithium carbonate, the optimal dosing should be governed by blood levels. Likewise, when prescribing or refilling clozapine, blood tests to monitor possible or probable neutropenia should be advised.

## Figures and Tables

**Figure 1 jcm-09-02624-f001:**
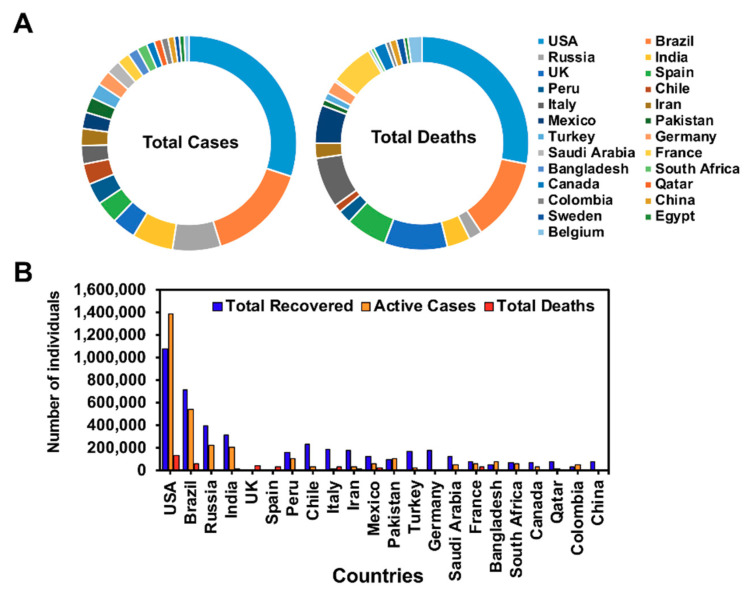
Total number of COVID-19 cases worldwide (25 countries on the list). Total confirmed cases and total deaths based on the countries (**A**), graphical representation of total recovered active cases, and fatalities based on the top 25 countries (**B**).

**Figure 2 jcm-09-02624-f002:**
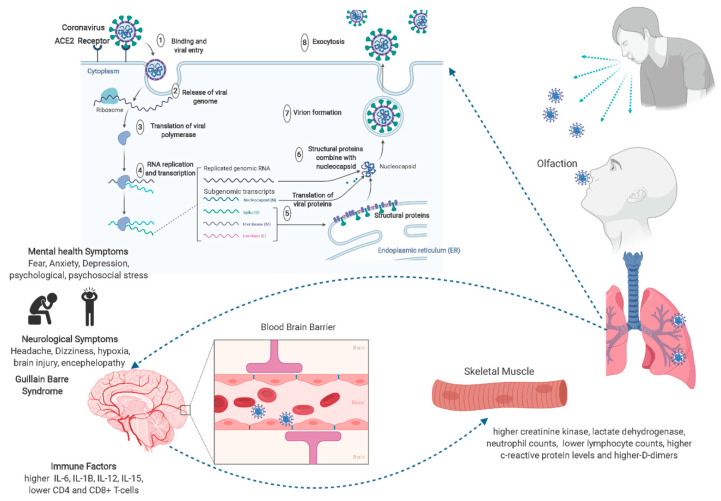
Putative mechanisms underlying neurobiological and psychological events of COVID-19 and their effect on mental health-related issues through psychoneuroimmunity. Virus–host interactions affect viral entry and replication—left panel: viral factor. SARS-CoV-2 is an enveloped positive single-stranded R.N.A. coronavirus. Two-thirds of viral R.N.A. is located in the first open reading frame that encodes 16 non-structure proteins. Host factors (right panel) can also influence susceptibility to infection and disease progression. COV2 enters the brain through olfaction, and since the virus is known to cross the blood–brain barrier, it can cause neurological symptoms like Guillain–Barré syndrome and mental health issues, including fear and anxiety for recovery. All these events are regulated by the cytokines and interleukins within the immune system.

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
