# Peer review of "The Interrelation of Neurological and Psychological Symptoms of COVID-19: Risks and Remedies"

_jcm, 2020, doi:10.3390/jcm9082624_

Round 1
Reviewer 1 Report
The present manuscript entitled „Neuropsychological Impact from COVID-19: Risks and Remedies” summarizes different aspects of psychological, neurobiological and neuroimmunological challenges and symptoms accompanying the COVID-19 pandemic.
The manuscript addresses a topic that may be of great interest to the readers of the Journal of Clinical Medicine. I would also like to acknowledge the importance of work in this area. However, I also noted substantial problems with the clarity of aims, target group, structure and language style that limit the potential for making a contribution in its present form. Although some of the conclusions represent important public and clinical health messages, most of the collected and compiled information are not novel and already published in even better forms of systematic reviews such as on the psychological impact of the COVID-19 pandemic. A more precise and detailed focus on the association between psychological and neurological symptoms related to COVID-19 infection, on the contrary, would have a greater chance to add value to the present knowledge. Thus, I would recommend reducing the scope and elaborating on interrelation in more detail, paying particular attention to clear and concise aims and keep focus throughout the manuscript. I have outlined more detailed comments below.
Title
- I recommend reconsidering the title: is this manuscript really about neuropsychological impact or about the interrelation between neurological and psychological symptoms of the COVID-19 pandemic? Moreover, “impact” suggests causal relationships and therefore requests longitudinal results and the thorough exclusion of confounding
Introduction
- What do you mean by “underlying mental illness” (p.2, l.48), the term seems to be inappropriate
- Your citation is inconsistent, for instance, “A. Cosini, 2020”
- I highly recommend to avoid generalized terms such as “no one” ,“everyone” or “we” (for general population), this is not accepted as good scientific standard (this pertains to the whole manuscript)
- The connection between mental health and athletics comes all of a sudden and needs to be derived from theory/previous studies more appropriately; moreover, top athletes are a very specific group and cannot be interpreted as representative for the general population
- The list of possible psychosocial challenges during a pandemic situation is correct but lacks reference to previous evidence
- In sum, after reading the introduction it remains unclear what this text is about and needs to be clarified; neuropsychological aspects are not even mentioned
Key challenges
- The authors refer to a large number of different mental disorders as well as symptoms without explanation, specification or clarification how they relate to each other to the focus of the present article
- The target groups differ unsystematically (i.e. health care professionals, patients; individuals with mental health problems,…)
- The same applies to the sudden listing of various health aspects and behaviours (“affective health, cognitive fitness, sleeping, and eating patterns,…, medication,…,hygiene”) – in how far is this important and related to the aim of your manuscript?
- Unclear why the authors display the worldwide number of COVID-19 infections
Advisable Remedies
- Please describe in how far your recommendations add value to the existing ones (e.g., as compared to those of the CDC, WHO)
- Please relate your recommendations to the referred results and structure analogous to the respective health categories and target groups
Conclusions
- What is an appropriate future perspective based on your results regarding research, clinical practice and everyday practice? For instance, I would expect more detailed information on how to implement your findings in the treatment of COVID-19 infections, especially in a more integral manner (taking neurological and psychological aspects into account)
- Be more careful with your subjective ethical and political recommendations – this is not in line with good scientific standard
Author Response
We thank the reviewers for their careful and constructive review of our manuscript. We were pleased to see that our review was received positively by the reviewers. The reviewer’s comments have been extraordinarily helpful in preparing a revised version of the manuscript, which we feel has been improved considerably through the review process. We carefully considered all of the comments, insights, and suggestions from both reviewers. We have addressed them in their entirety and edited the manuscript for any grammatical errors, and revisions to the text.
We hope that the reviewers find our rebuttal to suitably address their concerns.
- Reviewers’ comments: blue text
- Our responses: black regular text
- Sample of text from revised manuscript = red text.
Reviewer 1
The present manuscript entitled “Neuropsychological Impact from COVID-19: Risks and Remedies” summarizes different aspects of psychological, neurobiological and neuroimmunological challenges and symptoms accompanying the COVID-19 pandemic.
The manuscript addresses a topic that may be of great interest to the readers of the Journal of Clinical Medicine. I would also like to acknowledge the importance of work in this area. However, I also noted substantial problems with the clarity of aims, target group, structure and language style that limit the potential for making a contribution in its present form. Although some of the conclusions represent important public and clinical health messages, most of the collected and compiled information are not novel and already published in even better forms of systematic reviews such as on the psychological impact of the COVID-19 pandemic. A more precise and detailed focus on the association between psychological and neurological symptoms related to COVID-19 infection, on the contrary, would have a greater chance to add value to the present knowledge. Thus, I would recommend reducing the scope and elaborating on interrelation in more detail, paying particular attention to clear and concise aims and keep focus throughout the manuscript. I have outlined more detailed comments below.
Comment 1: Title: I recommend reconsidering the title: is this manuscript really about neuropsychological impact or about the interrelation between neurological and psychological symptoms of the COVID-19 pandemic? Moreover, “impact” suggests causal relationships and therefore requests longitudinal results and the thorough exclusion of confounding
Response: The reviewer’s comments is well taken, we have changes the manuscript title as suggested by the reviewer to “The interrelation between neurological and psychological symptoms of the COVID-19 pandemic: risks and remedies”
Comment 2: Introduction: What do you mean by “underlying mental illness” (p.2, l.48), the term seems to be inappropriate, Your citation is inconsistent, for instance, “A. Cosini, 2020”
Response: Thank for your this important observation, we have made the following changes to “previous history of mental illness” and deleted “underlying mental illness”. The reference A. Cosini has been deleted, since it was inconsistent with the flow of the manuscript as the reviewer suggested.
Comment 3: I highly recommend to avoid generalized terms such as “no one” ,“everyone” or “we” (for general population), this is not accepted as good scientific standard (this pertains to the whole manuscript).
Response: These generalized phrases such as “no one” ,“everyone” or “we” have been removed from the revised manuscript and scientific standard that is pertinent to the manuscript has been incorporated.
Comment 4: The connection between mental health and athletics comes all of a sudden and needs to be derived from theory/previous studies more appropriately; moreover, top athletes are a very specific group and cannot be interpreted as representative for the general population
Response: We thank the reviewer for pointing this out: the connection between mental health and athletes was abruptly quoted before. We have deleted this and represented the manuscript focused solely on the COVID-19 neurological and psychological effects in this population.
Comment 5: The list of possible psychosocial challenges during a pandemic situation is correct but lacks reference to previous evidence In sum, after reading the introduction it remains unclear what this text is about and needs to be clarified; neuropsychological aspects are not even mentioned.
Response: After reading the previous version of introduction section of the manuscript the focus was misleading, we tried to make the text clear and incorporated a new paragraph to maintain the flow of the manuscript. The new paragraph in the manuscript is as follows, also we included new references that goes with the work cited.
”In the latest pandemic time, it has become apparent that neurological and psychological involvement in COVID-19, are increasingly rising in the patient population. A subset of COVID patients is affected by neurological events such as headache, dizziness, or cerebrovascular symptoms [4-7]. The latest reports have also shown a sudden onset of anosmia and ageusia as an early sign of SARS-CoV-2 infection, suggesting that early neurological involvement might be highly relevant [4,5,7]. Currently, long-term neurological complications from COVID-19 infection are of great concern. Besides, the overarching issue is the situation of socio-economic crisis and psychological distress rapidly occurred worldwide. Although social activities have been restricted in most countries, almost all non essential movements were prohibited due to quarantine, leading to many psychological problems. The most important consequences of COVID-19 epidemic are of increased mental health issues, including stress, anxiety, panic symptoms, depression that has emerged increasingly [6,8]. The most common psychological symptoms are generalized fear and pervasive community anxiety, typically associated with disease outbreaks, with the escalation of new cases leading to increased anxiety and panic symptoms [6,8]. Overall, In this present review, we synthesize the literature to highlight the important associations between SARS-CoV-2 infection and the nervous system, mental health illness, discuss potential mechanisms of neural injury through psychoneuroimmunity. Awareness of the possible psychoneurological manifestations in COVID-19 patients is of utmost importance for management of potentially life-threatening and long lasting psycho-neurological complications.”
Comment 6 : Key challenges-The authors refer to a large number of different mental disorders as well as symptoms without explanation, specification or clarification how they relate to each other to the focus of the present article. The target groups differ unsystematically (i.e. health care professionals, patients; individuals with mental health problems,…) The same applies to the sudden listing of various health aspects and behaviours (“affective health, cognitive fitness, sleeping, and eating patterns,…, medication,…,hygiene”) – in how far is this important and related to the aim of your manuscript?
Response: We thank the reviewers for this comment- we tried to focus the review geared only towards interrelation between neurological and psychological symptoms from COVID-19 Pandemic, we tried to delete the health care professionals and all the general suggestions from the advisable remedies section. We have now currently focused on the how to deal psychological symptoms by coping, change of life style and having psychosocial counselling with the use of telemedicine to keep the flow the manuscript.
Comment 7: Unclear why the authors display the worldwide number of COVID-19 infections.
Response: With the increase in the number of cases worldwide, and the varying fatality rate in different countries, it is important to understand that with the increase in cases, it is highly important to screen the number of mental health and neurological cases as well, what are the longterm effects.
Comment 8: Advisable Remedies: Please describe in how far your recommendations add value to the existing ones (e.g., as compared to those of the CDC, WHO), Please relate your recommendations to the referred results and structure analogous to the respective health categories and target groups. Conclusions: What is an appropriate future perspective based on your results regarding research, clinical practice and everyday practice? For instance, I would expect more detailed information on how to implement your findings in the treatment of COVID-19 infections, especially in a more integral manner (taking neurological and psychological aspects into account), Be more careful with your subjective ethical and political recommendations – this is not in line with good scientific standard.
Response: We thank the reviewers for this comment- we have deleted the recommendations that could be subjective ethical and policital recommendations, tried to focus the review only to the scientific content of the manuscript with the interrelation between neurological and psychological symptoms from COVID-19 Pandemic. We have deleted the recommendations and the Helping the Helpers for the healthcare workers and the wellness groups, since it was deviating from the aims of the manuscript. We have now currently focused on the how to deal psychological symptoms by coping, change of life style and having psychosocial counselling with the use of telemedicine to keep the flow the manuscript.
Reviewer 2 Report
This is an interesting paper attempting to highlight current challenges that stem from the COVID-19 pandemic and to offer advice concerning coping with its psychological and psychosocial implications.
While some parts of the manuscript are very well written and comprehensible (sections 4 and 5), I would suggest overall careful revision of language and style, e.g.:
- congruent use of terms (“COVID-19 positive patients” or “COVID-19 patients;” I would suggest you use “COVID-19 patients”)
- please check for passages of repetition (e.g. 350: “The mental health care system should be persistent to persistent to provide...”; 398-400: “...there are an increasing number of studies that have reported pscyhoneurological symptoms. Certain studies have shown that one-third of COVID-19 patients show psychoneurological symptoms.”; 216: “...revealed an increase in a significant increase in...”
- please check for spelling errors and missing or extensive blanks (e.g. 399 and 404: “pscyhoneurological”, 112, 118, 152, 187, 189, 193, 224, 233, 272, 281, 285, 291, 295)
- please check for grammar/style (e.g. 46, 48-49, 69-70, 158, 172, 179: “consequences have led”, 185 “complained about” and “more severely ill patients”, 216: “Wuhan, China”)
- please check for unfinished sentences (e.g. 364, 367-370)
In the introduction, I’m not sure that I get the “athletes argument”. Why do athletes contracting the virus demonstrate the severity of the virus? Also, I think it is just normal that mental health is decreasing when one is no longer able to perform their job. Maybe skip that first passage.
In the section “Key challenges” you refer to hospital infrastructure that “had fallen apart” (77), please specify where (area) and when.
There are some parts of the paper where I feel it gets emotionally heated, I would recommend revision of these parts, toning them down (e.g. 53: “athletes who are idolized by their fans” à I do not see how this information/criticism is vital to the paper, 107: “we oddly see that common-sense is often being uncommon”, 410: “...instead of making assumptions...” à Who are you even criticizing here?
Moreover, I would recommend adding a couple of sources or references where needed (e.g. 91-93, 97-98 à Who recommends those self-care measures?, 300-303 (or is this already included in Kim&Su, 2020?), 326-347: “there are many tips and advice in the media...” à Could you back this up? Are you summarizing in this passage? Which “advice in the media” are you relating to?, 399-400 “Certain studies...” à Which ones?
Also, in some areas I find your recommendations rather vague. Can you elaborate on your call for “wellness modules” (374) or “wellness groups” (379)? à How should these look like? What should they include? How should they be made accessible? Who could develop them?; What does a “healthy diet” (342) look like?
Finally, in your conclusion you demand easier access to “many psychiatric medicines” (416-420). I feel that you didn’t build an argument here. Which compounds are you referring to? And why do you feel that prescription drugs should be made more easily available? If you are referring to antidepressants, they usually take a couple of weeks before potential improvement can be seen. If you are referring to anxiolytic compounds: Some of those (e.g. benzodiazepines) are highly addictive. There are existing guidelines when it comes to psychiatric prescription drugs. Please elaborate on which compounds you would recommend easier access to and why.
Author Response
We thank the reviewers for their careful and constructive review of our manuscript. We were pleased to see that our review was received positively by the reviewers. The reviewer’s comments have been extraordinarily helpful in preparing a revised version of the manuscript, which we feel has been improved considerably through the review process. We carefully considered all of the comments, insights, and suggestions from both reviewers. We have addressed them in their entirety and edited the manuscript for any grammatical errors, and revisions to the text.
We hope that the reviewers find our rebuttal to suitably address their concerns.
- Reviewers’ comments: blue text
- Our responses: black regular text
-Sample of text from revised manuscript = red text.
Comment 1: This is an interesting paper attempting to highlight current challenges that stem from the COVID-19 pandemic and to offer advice concerning coping with its psychological and psychosocial implications.
While some parts of the manuscript are very well written and comprehensible (sections 4 and 5), I would suggest overall careful revision of language and style, e.g.: Congruent use of terms (“COVID-19 positive patients” or “COVID-19 patients;” I would suggest you use “COVID-19 patients”)
Response: We thank the reviewer # 2 for all the productive comments, As per the suggestion, we have tried to maintain COVID-19 patients throughout the manuscript.
Comment 2: Please check for passages of repetition (e.g. 350: “The mental health care system should be persistent to persistent to provide...”; 398-400: “...there are an increasing number of studies that have reported pscyhoneurological symptoms. Certain studies have shown that one-third of COVID-19 patients show psychoneurological symptoms.”; 216: “...revealed an increase in a significant increase in...”
Response: We thank the reviewer for this observation, we have tried to delete the repetiion of words in the revised manuscript.
Comment 3: Please check for spelling errors and missing or extensive blanks (e.g. 399 and 404: “pscyhoneurological”, 112, 118, 152, 187, 189, 193, 224, 233, 272, 281, 285, 291, 295), Please check for grammar/style (e.g. 46, 48-49, 69-70, 158, 172, 179: “consequences have led”, 185 “complained about” and “more severely ill patients”, 216: “Wuhan, China”), Please check for unfinished sentences (e.g. 364, 367-370).
Response: We have checked the mansuscript for the grammatical errors and maintained the journal format in the revised version with the missing or extensive blanks.
Comment 4: In the introduction, I’m not sure that I get the “athletes argument”. Why do athletes contracting the virus demonstrate the severity of the virus? Also, I think it is just normal that mental health is decreasing when one is no longer able to perform their job. Maybe skip that first passage. There are some parts of the paper where I feel it gets emotionally heated, I would recommend revision of these parts, toning them down (e.g. 53: “athletes who are idolized by their fans” à I do not see how this information/criticism is vital to the paper,
Response: We thank the reviewer for pointing this out: the connection between mental health and athletes was abruptly quoted before. We have deleted this and represented the manuscript focused solely on the COVID-19 neurological and psychological effects in this population.
Comment 5: In the section “Key challenges” you refer to hospital infrastructure that “had fallen apart” (77), please specify where (area) and when.
Response: We thank the reviewer for this observation- We tried to address this as “Increased number of COVID-19 patients and shortage of staff and limited number of social interactions with the physician or healthcare professionals [11] led to a decline in face-to-face consultations and an increase in telemedicine” in the revised manuscript instead of quoting it as hospital infrastructure had fallen apart.
Comment 6: 107: “we oddly see that common-sense is often being uncommon”, 410: “...instead of making assumptions...” à Who are you even criticizing here?
Response: We thank the reviewer for the comment, we have deleted this general phrase ““we oddly see that common-sense is often being uncommon” from the revised version.
Comment 7: Moreover, I would recommend adding a couple of sources or references where needed (e.g. 91-93, 97-98 à Who recommends those self-care measures?, 300-303 (or is this already included in Kim&Su, 2020?), 326-347: “there are many tips and advice in the media...” à Could you back this up? Are you summarizing in this passage? Which “advice in the media” are you relating to?, 399-400 “Certain studies...” à Which ones?
Response: We thank the reviewers for this comment- We thank the reviewers for this comment- we have deleted the recommendations that could be subjective ethical, and policital recommendations, tried to focus the review only to the scientific content of the manuscript with the interrelation between neurological and psychological symptoms from COVID-19 Pandemic. As the reviewer pointed out most of the self-care measures and other tips are already in the social platform. Hence, we tried to delete the health care professionals and all the general suggestions from the advisable remedies section. We have now currently focused on the how to deal psychological symptoms by coping, change of life style and having psychosocial counselling with the use of telemedicine to keep the flow the manuscript.
Comment 8: Also, in some areas I find your recommendations rather vague. Can you elaborate on your call for “wellness modules” (374) or “wellness groups” (379)? à How should these look like? What should they include? How should they be made accessible? Who could develop them?; What does a “healthy diet” (342) look like?
Response: We have deleted the recommendations and the Helping the Helpers for the healthcare workers and the wellness groups, since it was deviating from the aims of the manuscript. We have now currently focused on the how to deal psychological symptoms by coping, change of life style and having psychosocial counselling with the use of telemedicine to keep the flow the manuscript.
Comment 9: Finally, in your conclusion you demand easier access to “many psychiatric medicines” (416-420). I feel that you didn’t build an argument here. Which compounds are you referring to? And why do you feel that prescription drugs should be made more easily available? If you are referring to antidepressants, they usually take a couple of weeks before potential improvement can be seen. If you are referring to anxiolytic compounds: Some of those (e.g. benzodiazepines) are highly addictive. There are existing guidelines when it comes to psychiatric prescription drugs. Please elaborate on which compounds you would recommend easier access to and why.
Response: We thank the reviewers for this comments. To address this we have incorporated the following in the revised version of the manuscript, at the end of conclusion”
“Increased screening of mental health symptoms during primary care or other medical specialty visits, and offering mental health services with the slightest degree of suspicion, can decrease the mental health burden. In most cases, the COVID -19 related stress in a person without significant past psychiatric history can be diagnosed as an adjustment disorder. Adjustment disorder can be treated with counseling and minimal use of psychotropic medications. On the other hand, patients with pre-existing or chronic psychiatric illnesses may decompensate due to increased stress related to COVID-19. This patient population may benefit from more frequent mental health care visits, even if this can be provided via telemedicine. Increased visits with their mental health provider can ensure medication compliance and early detection of any relapse of their psychiatric symptoms. In specific cases, the physician may consider more liberal but very brief use of anxiety and sleep medications. The decision to use medication should be based upon the most recent evidence-based guidelines.
An accumulating body of recent evidence proposes that anxiety, depressive and psychotic symptoms are all likely to worsen during extreme COVID-induced stress and social disruption. Moreover, patients will be at increased risk of relapse or recurrence of affective and psychotic illness. As such, it will be important when deciding on the best management plan (non-pharmacological/pharmacological) to consider all the relevant factors, including risk to self and others. It is important to understand the difference between short and long term use of psychotropic medications, and also clarify the myths related to the addiction potential of all psychotropic. Although many psychiatric medicines are tightly regulated, and as prescribed-only for long-term mental illnesses, it may be necessary for the governments to ease up prescription refill regulations. Tele-therapy and online consultations with e-prescriptions would allow ease of access to the prescribed medications without referring to the mental healthcare provider in person. Perhaps in many occurrences, designated pharmacies might collect e-prescriptions plus the related contact info of the patients to home-delivery medicines. Notwithstanding the above, possible untoward or side-effects of such medication need to be noted. In other words, upon treatment with commonly prescribed psychotropic drugs, careful consideration should be given to whether now is the best time to commence, withdraw, or change patients from an antidepressant, anxiolytic or antipsychotic medications. For instance, in patients who receive ongoing treatment with benzodiazepines, the potential for tolerance and dependence need to be considered. With regards to lithium carbonate, the optimal dosing should be governed by blood levels. Likewise, when prescribing or refilling clozapine, blood tests to monitor possible or probable neutropenia should be advised”
In addiditon to all the reviewer’s comments, we have tried to made some changes in the abstract with the aims of the manuscript and we introduced a new paragraph on neurological and psychological symptoms of COVID-19 to maintain the flow of the manuscript.
Psychological and Neurological Symptoms of COVID-19:
The major psychological symptoms among people include signs of anxiety, panic attacks, depression, and suicide [19,20]. To elaborate, the symptoms include persistent worrying or feeling overwhelmed by emotions; restlessness and irritability; sleep problems like insomnia or excessive sleeping; sweating, trembling, shortness of breath or a sense of choking; lack of interest, significant weight loss/gain, feelings of worthlessness or excessive guilt, repeated thoughts of death or suicide [6,19,20]. In a latest resport from Ellul et al., (2020), observed patients with COVID-19 with neurological or psychiatric disease reported over three weeks, majority of them had altered mental status, which included encephalopathy, with a neuropsychiatric diagnosis, including psychosis, neurocognitive (dementia-like) syndrome, and affective disorders[7]. Notably, patients had a cerebrovascular event, ischaemic strokes, intracerebral hemorrhages, CNS vasculitis, and other cerebrovascular events[6,7]. Hence it is highly important to understand the underlying biological mechanism on how these psychoneurological events are affected from COVID-19.
Round 2
Reviewer 1 Report
The authors have clarified and addressed several of the questions I raised in my previous review so that the manuscript has the potential to be accepted in its current form. However, I still noticed grammatical inconsistencies that may require further thorough editing.